# POINTWORLD: SCALING 3D WORLD MODELS FOR IN-THE-WILD ROBOTIC MANIPULATION

## ABSTRACT

Humans anticipate, from a glance and a contemplated action of their bodies, how the 3D world will respond. This predictive ability is equally vital for enabling robots to manipulate and interact with the physical world. We introduce PointWorld, a foundation 3D world model that unifies state and action in a shared spatial domain and predicts 3D point flow over short horizons: given one or a few RGB-D images and a sequence of robot actions, PointWorld forecasts per-point scene displacements that responds to the actions. To train our 3D world model, we curate a large-scale dataset for 3D dynamics learning spanning real and simulated robotic manipulation in diverse open-world environments—enabled by recent advances in 3D vision and diverse simulated environments—totaling about 2M trajectories and 500 hours. Through rigorous, large-scale empirical studies of backbones, action representations, learning objectives, data mixtures, domain transfers, and scaling, we distill design principles for large-scale 3D world modeling. PointWorld enables zero-shot simulation from in-the-wild RGB-D captures. It also powers model-based planning and control on real hardware that generalizes across diverse objects, and environments, all without task-specific demonstrations or training. We will release all the code and data. For more videos, please check: https://sites.google.com/view/pointworld

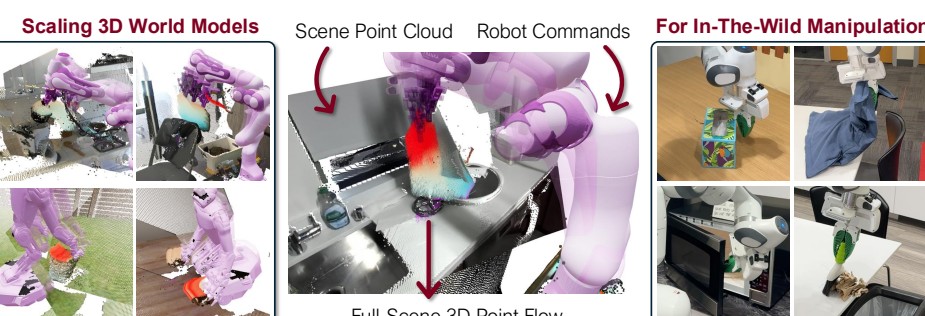

Figure 1: **PointWorld overview.** We curate a large-scale 3D world modeling dataset for robotic manipulation and scale up training for 3D world models. Pretrained on diverse data, a single model is capable of predicting dynamics in open-world environments zero-shot to guide robotic manipulation on rigid-body pushing, deformable, articulated manipulation, and tool use.

## 1 INTRODUCTION

World modeling in unstructured environments is imperative for general-purpose robots: predicting how the world will evolve from what the robot sees and intends to do with its body. Humans do this from a glance and a grasp—forecasting deformation, articulation, stability, and contact—revealing how much a world-modeling objective captures when conditioned on a contemplated action in 3D. Actions unfold where physics lives, in space and time: our aim is a predictive model that makes such spatially grounded, action-conditioned predictions from only perceptual inputs in open-world settings.

A large body of work has studied world modeling from complementary angles. Physics-based models (Todorov et al., 2012; Coumans, 2015; Makoviychuk et al., 2021; Hu et al., 2020) are capable of making highly accurate predictions, but face sim-to-real gaps and require curated, environment-specific modeling. Learning-based dynamics models (Sanchez-Gonzalez et al., 2020; Pfaff et al., 2020; Huang et al., 2025; Janner et al., 2019; Chua et al., 2018) address this by learning from observed interaction, yet often depend on domain-specific inductive bias (e.g., full observability, objectness priors, or material specification), which constrains scaling and contrasts with human-level generalization. In parallel, large video generative models trained at scale (Finn et al., 2016; Oh et al., 2015; Li et al., 2025; Bardes et al., 2024; Agarwal et al., 2025; Yang et al., 2023; 2025) are capable of producing photorealistic predictions but lack explicit action conditioning and often fall short on physical consistency. For a recent synthesis of learning-based dynamics models in manipulation, see Ai et al. (2025). Despite progress, a gap remains between what current models predict and what humans can foresee from visual observations and a contemplated action in 3D.

Our philosophy is unification for scaling: represent *state* and *action* in the same natural modality of 3D physical space. State is represented by a full-scene 3D point cloud built from RGB-D captures; actions are dense 3D point trajectories instantiated from the agent's embodiment—typically known a priori (e.g., a robot description)—and thus forecastable over time. Under this representation, 3D world modeling equates to modeling *full-scene 3D point flow* under perturbations from a temporal sequence of robot points: given partially observed 3D scene points and those action points, predict per-point scene displacements over the horizon. While conceptually simple, this formulation ties raw sensory observation and an embodiment-agnostic action space in a shared representation through dynamics (what moves, how, and where) and implicitly captures objectness, articulation, and material properties, all through interaction with the robot's specific geometry (e.g., grippers, fingers) from partial observations. Unifying everything in 3D enables scaling across embodiments and heterogeneous data, akin to "next-token prediction" (Brown et al., 2020) but for interaction over 3D space and time. We term our approach as **PointWorld**.

To provide supervision, we curate the first large-scale dataset for 3D dynamics modeling for manipulation, spanning more than 2000 in-the-wild scenes with single-arm, bimanual, and whole-body interactions across both real and simulated domains. The dataset was built from existing robotic manipulation datasets (DROID (Khazatsky et al., 2024) and BEHAVIOR-1K (Li et al., 2024)), with our custom high-quality 3D annotations enabled by recent advances in 3D vision (metric depth estimation (Wen et al., 2025), calibration (Wang et al., 2025), tracking (Karaev et al., 2023)). Leveraging the dataset, we train a *foundation 3D world model* that conditions on 3D action flow and learns with a correspondence objective. We distill important design decisions for large-scale 3D dynamics modeling through rigorous and comprehensive investigations of backbone architectures, action representations, objectives, data mixtures, scaling laws, and domain transfers under both zero-shot and finetuned settings.

To demonstrate its potential for robotic manipulation, we employ the pre-trained model in a model-predictive control (MPC) framework for action inference with a real robot. As PointWorld predicts scene dynamics jointly over short action chunks, it provides a natural and efficient integration with sampling-based MPC (e.g., MPPI (Williams et al., 2017)). Pre-trained on diverse interactions, the same model is deployed zero-shot in the wild for manipulation tasks involving diverse objects and generalizing to previously unseen embodiments. This shows how a unified 3D dynamics model can power practical manipulation through model-based planning and control, without relying on task-specific training or demonstrations.

**Contributions.** 1) We introduce a foundation 3D world model, PointWorld, that unifies state and action in a common spatial representation and predicts point flow from RGB-D observations and robot action inputs; through rigorous, large-scale studies we distill a general recipe for important design choices. 2) We open-source the first large-scale dataset for 3D dynamics modeling ($\sim$ 2M trajectories, $\sim$ 500 hours) sourced from existing robotics datasets with our custom high-precision 3D annotations. 3) Leveraging the pretrained model, we demonstrate model-based planning and control on real hardware operating zero-shot in the wild across diverse objects and environments, with generalization to previously unseen embodiments. Through these studies, we hope to demonstrate the potential of 3D world modeling for robotic manipulation and lay the foundation for future advancement.

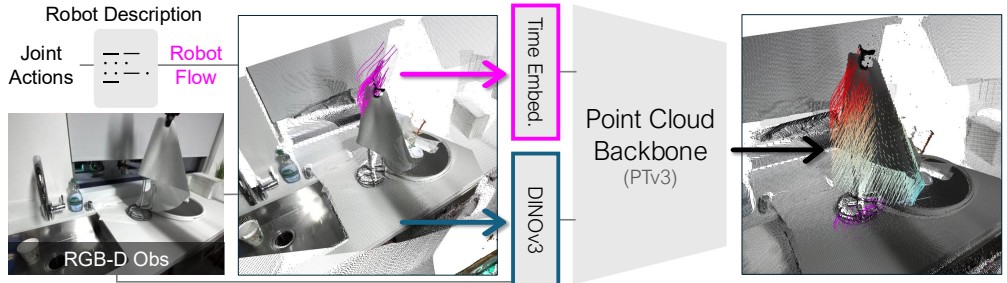

Figure 2: **Overview of PointWorld**. PointWorld fuses RGB-D observations into a unified point cloud and augments it with dense robot trajectories and predict full-scene 3D point flow.

## 2 RELATED WORK

A world model or dynamic model is an engineered or learned transition map from current state and action to next state. With a world model, one can leverage several model-based methods like online model-predictive control or offline policy optimization to determine the appropriate robot actions to tackle a given robotic manipulation task. A substantial body of research has explored world modeling from various complementary perspectives. Physics-based models (Todorov et al., 2012; Coumans, 2015; Tedrake & the Drake Development Team, 2019; Makoviychuk et al., 2021; Koenig & Howard, 2004; Hu et al., 2018; 2020; Huang et al., 2021; Sulsky et al., 1993) can deliver highly accurate predictions but often suffer from sim-to-real transfer issues and require carefully crafted, environment-specific designs. Learning-based dynamics models (Battaglia et al., 2016; Sanchez-Gonzalez et al., 2020; Pfaff et al., 2020; Huang et al., 2025; Ha & Schmidhuber, 2018; Hafner et al., 2018; 2019; 2020; Sekar et al., 2020; Janner et al., 2019; Chua et al., 2018) learn directly from observed interactions, mitigating some of these challenges. However, they typically rely on strong domain-specific inductive biases—such as full observability, object-centric representations, or predefined material properties—which can hinder scalability and fall short of the generalization seen in humans. Large-scale video generative models (Finn et al., 2016; Lee et al., 2018; Oh et al., 2015; Lotter et al., 2016; Denton & Fergus, 2018; Yan et al., 2021; Ho et al., 2022b; Singer et al., 2022; Ho et al., 2022a; Guo et al., 2023; Yin et al., 2023; Chen et al., 2023; Li et al., 2025; Chen et al., 2024; Song et al., 2025; Hong et al., 2024; Bai et al., 2025; Bardes et al., 2024; Assran et al., 2025; Agarwal et al., 2025; Yang et al., 2023; 2025) can produce photorealistic future frames but usually lack explicit action conditioning and struggle with maintaining physical plausibility. For a recent overview of learning-based dynamics models in the context of manipulation, see Ai et al. (2025). In this work, we aim to develop a foundational 3D world model using large-scale real-world data across diverse tasks, enabling zero-shot deployment in the wild on real robots with varying embodiments.

## 3 METHOD

Herein we first formulate 3D world modeling as action-conditioned 3D point flow prediction under minimal assumptions (Section 3.1). We then instantiate this with PointWorld, a model that predicts full-scene 3D point flow conditioned on a temporal sequence of robot action points derived from the embodiment (Section 3.2). Finally, we describe how to integrate the learned dynamics into model predictive control for robotic manipulation tasks (Section 3.3).

### 3.1 PROBLEM FORMULATION

We consider a dataset with: (i) calibrated RGB–D fused into a partial point cloud per time $t$; (ii) a robot description; and (iii) step-to-step correspondences (for supervision). At test time, the model consumes current RGB–D, robot joint configurations over a short horizon, and the robot description—no correspondence is needed. The state is a 3D point cloud $\mathbf{S}_t = \mathbf{P}_t \in \mathbb{R}^{N \times 3}$ and we note the robot actions as $\mathbf{u}_t$. The one-step dynamics are a learned function of state and action:

$$\mathbf{S}_{t+1} = \mathcal{F}_\theta(\mathbf{S}_t, \mathbf{u}_t), \tag{1}$$

where $\mathcal{F}_\theta$ is a learned transition function. We use relative prediction (Chi et al., 2024), outputting a per-point displacement $\Delta\mathbf{P}_{t\to t+1}$ added to the input:

$$\mathbf{P}_{t+1} = \mathbf{P}_t + \Delta\mathbf{P}_{t\to t+1}, \qquad (2)$$

explicitly preserving correspondences.

Points are (position, feature) tuples. Scene: $(\mathbf{p}_i, \mathbf{f}_i^S)$ with $\mathbf{p}_i \in \mathbb{R}^3$, $\mathbf{f}_i^S \in \mathbb{R}^{D_S}$ and $\mathbf{S}_t=\{(\mathbf{p}_i, \mathbf{f}_i^S)\}_{i=1}^N$. Robot: $(\mathbf{r}_j, \mathbf{f}_j^R)$ with $\mathbf{r}_j \in \mathbb{R}^3$, $\mathbf{f}_j^R \in \mathbb{R}^{D_R}$. Feature choices are detailed in Section 3.2.

Actions over the next $H$ steps are joint configurations $\{\mathbf{q}_{t+k}\}_{k=1}^H$. We sample robot points once at the first frame, attach them to corresponding links, and propagate them to each step via forward kinematics, yielding per-step robot point sets and preserving robot-point correspondences. For efficiency we keep only the gripper subset (see Section **??**). Training matches predicted and ground-truth displacements/positions where correspondences are valid.

## 3.2 3D WORLD MODELING WITH POINTWORLD

We instantiate Eq. equation 1 with a pipeline that builds a unified interaction point set from RGB–D and robot kinematics and predicts scene flow.

**Unified point set and feature streams.** We fuse $M$ calibrated RGB–D views into a partial scene point set in the world frame. Using known camera poses and robot configurations, we exclude robot pixels when forming scene points. In parallel, given joint configurations $\{\mathbf{q}_{t+k}\}_{k=1}^H$ and the robot description, we sample robot surface points once at the first frame, associate them with links, and propagate them to each future step by forward kinematics, preserving correspondences across steps. Scene points are featurized with a frozen DINOv3 ViT-L/16 (multi-scale aggregation via projection and view pooling; (Siméoni et al., 2025; Duisterhof et al., 2025)), while robot points are linearly projected and augmented with a sinusoidal temporal embedding and a learned robot-type embedding. The concatenation of scene points (at time $t$) and the temporally unfolded robot points forms a single unified point set processed by the backbone. This unification presents an embodiment-agnostic 3D state in which robot motion appears as a time-parameterized subset, enabling one model to scale across heterogeneous data and learn an action-conditioned scene-flow field.

**Backbone and heads.** A custom Point Transformer V3 (about 410M parameters; (Wu et al., 2023)) processes the unified point set and outputs features for all scene points, followed by: (i) a dynamics head that predicts per-step 3D point displacements over the horizon; and (ii) an aleatoric-uncertainty head that outputs a scalar log-variance $\{s_{t\to t+k}\}_{k=1}^H$ per point. Predicted positions are $\mathbf{P}_{t+k} = \mathbf{P}_t + \Delta\mathbf{P}_{t\to t+k}$. The overall capacity is roughly 700M parameters including the frozen DINOv3 encoder, which is beneficial for modeling fine-grained, spatial interactions in real scenes.

**Training Objective.** Most points are static; without reweighting, a model can minimize error by predicting near-zero flow. We weight per-point losses by soft movement likelihoods and attenuate noisy supervision via aleatoric uncertainty. Define movement weights inline from soft movement likelihoods $m_{k,i} \in [0,1]$ as $w_{k,i}^{\text{move}} = m_{k,i}/\sum_{k,i} m_{k,i}$. Writing the objective over absolute positions and summing over non-occluded, valid-depth correspondences, we minimize

$$\mathcal{L}_{\text{dyn}} = \tfrac{1}{2}\sum_{k=1}^H \sum_i w_{k,i}^{\text{move}} \left( \rho_\delta(\hat{\mathbf{P}}_{t+k,i} - \mathbf{P}_{t+k,i})\, e^{-s_{k,i}} + \lambda\, s_{k,i} \right). \qquad (3)$$

Here $\rho_\delta(\cdot)$ is the Huber function applied elementwise to the 3D residual and averaged across dimensions; $s_{k,i}$ is the scalar log-variance predicted by the uncertainty head; and $\lambda$ is a constant. In practice, the uncertainty term improves robustness to real-data noise without explicit labels. The 2D encoder is frozen. See Appendix A.1 for further details.

## 3.3 POINTWORLD FOR ROBOTIC MANIPULATION

We couple PointWorld with a sampling-based planner that proposes absolute end-effector pose trajectories (SE(3)). At test time the model needs only RGB-D snapshot(s), the robot's joint sequence over the horizon, and the robot description—correspondence is required only during training. Crucially,

| Sensor Depth + Manual Ext. | FS Depth + Optimized Ext. | Sensor Depth + Manual Ext. | FS Depth + Optimized Ext. |

Figure 3: Annotation pipeline. Our real-world DROID data pipeline couples learned stereo depth, mesh-constrained extrinsics refinement, and mask-guided point tracking to deliver dense, marker-free 3D trajectories that surpass raw sensor captures.

the dynamics predict the entire horizon in one feed-forward pass, enabling efficient evaluation of candidate trajectories.

Given an observation, we form the unified point set (Section 3.2). The learned dynamics roll out the horizon by chaining model windows to yield predicted scene evolution and end-effector poses. A sampling-based optimizer evaluates candidates and updates the nominal.

We separate task objectives from control regularization. We consider a task cost defined in the model's state space—namely, the 3D trajectories of scene points $\mathbf{S}_{t+1:t+H}$—which applies broadly across rigid, deformable, articulated, and tool-use manipulation; concrete instantiations appear in Experiments. Control cost uses an SE(3) path-length term and an IK reachability residual. The overall optimization is

$$\min_{\mathbf{u}_{t+1:t+H}} \sum_{k=1}^{H} \Big[ c_{\text{task}}\big(\mathbf{S}_{t+k}\big) \; + \; c_{\text{ctrl}}\big(\mathbf{u}_{t+k}, \mathbf{E}_{t+k}\big) \Big] \quad \text{s.t. } \mathbf{S}_{t+1:t+H} \text{ satisfies Eq. equation 1}, \quad (4)$$

where $\mathbf{E}_{t+k}$ is the end-effector pose at step $t+k$. Deployment specifics (robot embodiment, sensing, workspace/orientation bounds, sampling schedule, and cost settings) appear in Appendix A.1.

## 4 DATASET CURATION AND EVALUATION PROTOCOL

The world modeling objective in Eq. 1 targets open-world simulation where objects, hands, and cameras move with complex, contact-rich dynamics. Achieving reliable learning in this setting requires a large and diverse dataset with accurate 3D supervision. Historically, such supervision has been difficult to obtain in the real world. Our key observation is that recent advances in 3D vision—metric stereo depth estimation, robust multi-view calibration, and long-range point tracking—enable a fully markerless pipeline that yields high-quality 3D annotations without manual effort (Figure 3). Complementarily, modern photorealistic simulation provides perfect ground-truth supervision at scale. Building on both fronts, we curate a dataset of roughly $\sim 2$M trajectories ($\sim 500$ hours) spanning single-arm, bimanual, and whole-body manipulation across in-the-wild real scenes and simulated households.

**Real data (DROID).** We leverage the DROID dataset (Khazatsky et al., 2024), which logs stereo RGB and robot proprioception across diverse open-world manipulation. Raw captures are not directly suitable for precise 3D dynamics learning: sensor depth is noisy, dataset extrinsics can drift by centimeters, and there is no dense 3D correspondence. Accurate 3D is essential for generalization in the wild—the model must localize the hand relative to the scene to reason about contact and interaction. After extensive experimentation, we found that off-the-shelf methods alone (e.g., VGGT (Wang et al., 2025)) do not achieve the accuracy needed for manipulation purposes.

Our pipeline addresses these issues in three steps. First, we replace sensor depth with metric stereo using a strong learned estimator (Wen et al., 2025), which is particularly effective at the close working distances typical of manipulation. Second, we compute camera extrinsics from scratch by initializing with Wang et al. (2025) and then jointly refining per-camera 6-DoF poses by aligning the observed depth of the robot body to a known robot mesh, constrained by robot kinematics and proprioception. This "eye-in-hand/eye-on-base" refinement closes the remaining centimeter-level gap and does not

require measured initial extrinsics. Third, given accurate depth and extrinsics, we obtain per-pixel correspondences using modern point trackers (e.g., CoTracker (Karaev et al., 2023)), masking out robot and out-of-workspace regions. As a result, we recover reliable depth and extrinsics for over 60% of DROID (nearly 200 hours of human teleoperation), yielding what we believe is the largest real-world 3D manipulation dataset with dense scene trajectories. Qualitatively, our annotations substantially improve over raw sensor depth and dataset extrinsics (see Figure 3), which will be open-sourced to the community.

**Simulation (BEHAVIOR-1K).** To complement real-world data, we use BEHAVIOR-1K (Li et al., 2024), which provides $\sim 1200$ hours of teleoperated interaction in photorealistic household environments with bimanual, whole-body, and mobile manipulation. Simulation supplies perfect 3D supervision, covering long-horizon diverse activities in household scenes.

**Evaluation protocol.** We evaluate predicted point flow using a per-point per-timestep MSE over the prediction horizon. Because most scene points remain static, we focus on evaluating the moving points (as measured by gt data) by reusing the movement-aware weighting introduced in Section 3. For real data (DROID), to additionally address the remaining noise in the evaluation data, we train a separate expert model only on the held-out test split to flag unreliable flows via the uncertainty objective from Section 3. All other models are trained exclusively on the imperfect training data and are evaluated on the expert-filtered test annotations. We report dataset- and domain-level aggregates and include per-domain meta-statistics; further details appear in Appendix A.3.

## 5 EXPERIMENTS

### 5.1 SCALING 3D WORLD MODELS: A ROADMAP

**Overview.** Leveraging our carefully curated dataset, we perform a systematic study of the design choices that matter when scaling 3D world models for robot manipulation. To our knowledge, this is the first investigation into scaling up action-conditioned 3D world modeling by leveraging a diverse pretraining corpus of real and simulated robot interaction. Throughout, we note that absolute metric changes can appear modest because all errors are measured over one-second horizons where even large motions displace points by only a few centimetres. Empirically, the seemingly small numerical improvements correspond to pronounced jumps in rollouts and downstream planning, as illustrated in Figure 7.

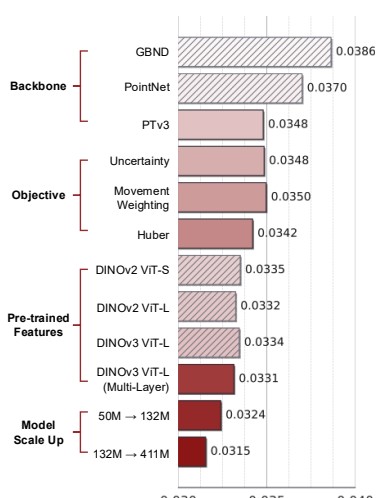

Figure 4: Roadmap overview. Cumulative gains from backbone, objective, features, and scale.

**Architecture.** Graph-based neural dynamics (GBND) models remain the default in learning-based physics modeling thanks to their expressive relational inductive bias Ai et al. (2025). We first scale a GBND baseline to our dataset and observe the challenges documented in Table 1: memory consumption grows rapidly even for modest capacity because the formulation attends to all fine-grained input points without higher-level abstraction; furthermore, the purely local message passing struggles under partial observability since long-range effects must traverse many noisy hops. Motivated by these limitations we study alternative point cloud encoders, progressing from PointNet, PointNet++, and sparse convolutional nets to transformer-based designs. Transformers, and in particular the PTv3 family, deliver consistently stronger modeling power while being far more memory efficient. The U-Net style hierarchy in PTv3 mirrors the strengths of GBND—local grouping and multi-scale abstraction—yet attention over progressively coarser point sets enables both long-range interaction modeling and substantial parameter growth. Table 1 shows that we can scale parameters by roughly $300\times$ (GBND to PTv3-411M) while keeping memory and runtime comparable, yielding a clear improvement in prediction accuracy. Together, these results motivate our choice of PTv3 as the default backbone.

**Training objective.** A naïve $\ell_2$ regression objective quickly collapses toward predicting no motion because manipuland points constitute only 1–5% of a scene with tens of thousands of points. We therefore reweight residuals according to ground-truth displacement magnitude, effectively applying weighted least squares to mitigate regression imbalance Hastie et al. (2009). Because real-world

| Backbone | Params | Mem. | FLOPs | $\ell_2$ mov. | $\ell_2$ stat. |
|---|---|---|---|---|---|
| ○ GBND | 1.00 | 1.00 | 1.00 | 0.0390 | 0.0066 |
| ○ PointNet | 1.02 | 0.26 | 0.02 | 0.0369 | 0.0084 |
| ○ PointNet++ | 1.05 | 0.68 | 0.05 | 0.0368 | 0.0073 |
| ○ SparseConv | 25.2 | 3.54 | 1.09 | 0.0396 | 0.0076 |
| ○ Transformer | 31.0 | 0.21 | 2.65 | 0.0339 | 0.0071 |
| ○ PTv3-50M | 36.9 | 0.19 | 0.34 | 0.0331 | 0.0067 |
| ○ PTv3-132M | 95.6 | 0.36 | 0.68 | 0.0324 | 0.0061 |
| ● PTv3-411M | **299.2** | 0.99 | 1.84 | **0.0315** | **0.0059** |

Table 1: Backbone comparisons on the DROID test set. Lower $\ell_2$ indicates more accurate scene reconstructions.

captures inevitably contain noise, we follow uncertainty-aware objectives such as VGG-T Wang et al. (2025) by predicting per-point aleatoric variance, and we replace $\ell_2$ with a Huber loss to limit the influence of outliers. All three changes act in concert: movement-based weighting alone would over-emphasise noisy signals, but the uncertainty head and robust loss temper the weights and prevent overfitting. The combined objective greatly stabilizes training and improves robustness across the diverse data mixture.

**Pretrained features.** Although geometric reasoning in 3D is compelling, high-quality pretrained 3D representations remain scarce; methods such as SONATA Wu et al. (2025) make encouraging progress yet currently lack the granularity required for manipulation-scale dynamics. We therefore project points into calibrated cameras and attach frozen 2D features from DINOv3 Siméoni et al. (2025). This simple addition meaningfully boosts accuracy and stabilizes optimization, likely because the dense features provide a prior over objectness and semantics without enforcing explicit segmentation. The trend echoes recent 3D perception systems that benefit from DINO-style dense descriptors Wen et al. (2025); Wang et al. (2025). Implementation details—including multi-view sampling and feature normalisation—are summarized in the appendix.

**Scaling parameters.** With architecture, objective, and features in place, we expand depth and width within the PTv3 blueprint, aligning with scaling-law observations in vision and language modeling Kaplan et al. (2020). Model sizes spanning 50M to 411M parameters exhibit a smooth, approximately log-linear gain on the DROID benchmark (Figure 6), reinforcing the importance of jointly scaling data and capacity.

**Summary.** Our roadmap shows that each lever—architecture, objective, features, and scale—contributes to a marked qualitative leap despite modest absolute metric shifts. Collectively these advances yield far sharper rollouts and stronger downstream planning, providing a solid foundation for future work on even richer 3D world modeling.

## 5.2 Ablations

**Action representation.** We model robot actions as dense point flows sampled from the manipulators, allowing the backbone to reason about state and action in a shared geometric space. For efficiency we currently render only the active effectors—roughly 300–500 points per time step in our datasets—and we compare against four baselines: (i) whole-body point clouds with the same number of points (sparser coverage), (ii) whole-body point clouds with 2000 points (similar density but $\approx 2\times$ higher compute at train and test time because each rollout aggregates $\sim 20,000$ robot points), (iii) a low-dimensional end-effector pose, and (iv) a low-dimensional joint configuration. Low-dimensional variants omit robot points entirely; we concatenate their signals with the scene features before the backbone. Each model is trained on the combined DROID and BEHAVIOR-1K dataset.

On the BEHAVIOR-1K benchmark, the results align with intuition: fusing robot state and action into the point cloud substantially improves mover accuracy relative to low-dimensional inputs, as the model can explicitly capture contact geometry. Sparse whole-body points underperform the gripper-only flow because they dilute attention over large rigid links and miss contact cues, while dense whole-body points help but still lag due to inefficient gradient propagation through inactive body parts. Rendering only gripper points delivers the best accuracy while remaining twice as efficient as the 2000-point alternative. In the noisier real-world DROID setting the picture is subtler: naively

|  |  | In-domain | | Cross-domain | | Held-out real | | | From |
| --- | --- | --- | --- | --- | --- | --- | --- | --- | --- |
|  |  | D → D | B → B | D → B | B → D | D → H | B → H | D+B → H | scratch |
| **Zero-shot** | $\ell_2$ mover | 0.0315 | 0.0087 | 0.1460 | 0.0558 | 0.0305 | 0.0531 | 0.0300 | 0.0293 |
|  | $\ell_2$ static | 0.0059 | 0.0010 | 0.0050 | 0.0058 | 0.0049 | 0.0057 | 0.0063 | 0.0043 |
| **Finetuned** | $\ell_2$ mover | – | 0.0107 | – | 0.0378 | 0.0271 | 0.0299 | 0.0272 | 0.0293 |
|  | $\ell_2$ static | – | 0.0003 | – | 0.0086 | 0.0040 | 0.0046 | 0.0040 | 0.0043 |

Table 2: Generalization and transfer summary. D denotes the DROID benchmark, B the BEHAVIOR-1K household suite, and H the real-world CLVR evaluation. The shaded rows mark static-scene accuracy and the final column compares against a model trained solely on the held-out scenes.

including entire robot bodies exacerbates supervision noise and can make low-dimensional baselines look competitive. However, the compact gripper-only representation retains the expressiveness needed for contact reasoning and ultimately surpasses the low-dimensional variants, underscoring the importance of matching action fidelity to data quality.

**Scaling analysis.** Inspired by scaling-law studies in language and vision Kaplan et al. (2020); Chowdhery et al. (2022); Alayrac et al. (2022), we examine whether our 3D world models exhibit similar trends. Focusing on the diverse DROID corpus, we vary model capacity (50M, 132M, and 411M parameter PTv3 variants) and data fraction (5%, 10%, 25%, 50%, 100%). Each curve sweeps one axis while keeping the other fixed. When plotted in log space we observe approximately linear behaviour for both axes (Figure 6), suggesting that additional data and capacity provide predictable gains. The observation that mixtures of simulated and real data follow the same slope indicates that the model leverages the breadth of experiences rather than overfitting to a single domain.

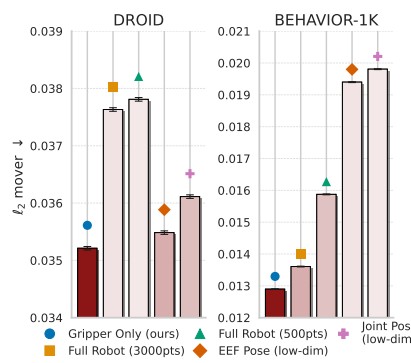

Figure 5: Action representation ablation.

## 5.3 GENERALIZATION AND TRANSFER

**Evaluation protocol.** We probe how far the pretrained model generalizes across domains, embodiments, and environments. Each finetuning experiment is restricted to one twentieth of the original training iterations, forcing the model to rely on priors learned from the large-scale mixture. In-domain studies use held-out splits of DROID and BEHAVIOR-1K; for DROID we follow the filtered evaluation protocol to avoid overlap with training scenes and report test metrics for reference.

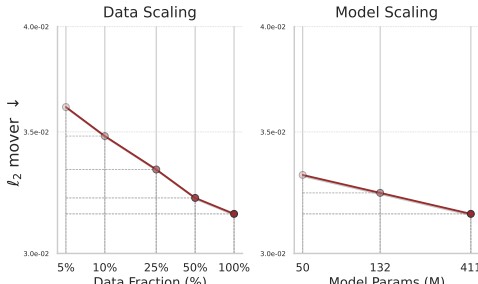

Figure 6: Scaling Study.

**In-domain.** On BEHAVIOR-1K (simulation with clean labels) the model achieves sub-centimetre mover error on held-out episodes, while DROID performance remains stable despite real-world sensor variation. These results indicate that our scaling recipe does not merely memorize training trajectories.

**Cross-domain.** Zero-shot transfer between simulation and real-world domains remains challenging, mirroring prior observations in manipulation. Nevertheless, finetuning with only 5% of the original optimization steps rapidly narrows the gap to domain-specific models trained from scratch with 20× more updates. The effect is symmetric: real-to-sim and sim-to-real transfers both benefit, suggesting that either source can provide useful priors for 3D world models once combined with a modest amount of target data.

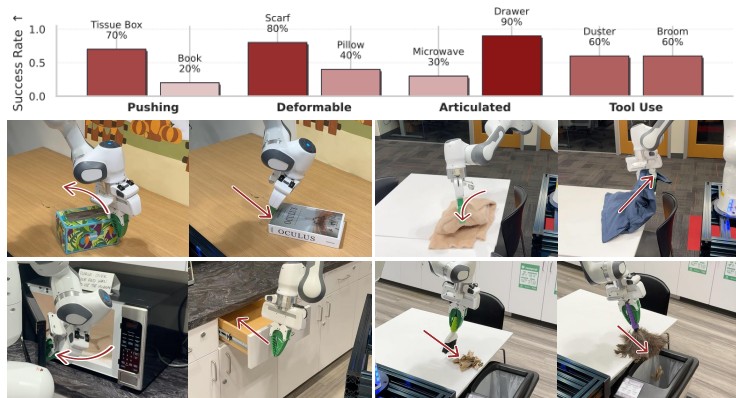

Figure 7: Model-based planning with PointWorld. Zero-shot deployment within MPC supports rigid pushing, deformable, articulated, and tool-use tasks on real hardware.

**Held-out environment.** To approximate an unseen real environment at scale, we hold out data from the geographically distant CLVR lab within DROID. The held-out set is split into 90% train and 10% test; zero-shot models never see these frames, while finetuned variants access only the 90% subset. Architectures pretrained on the remaining DROID sites achieve surprisingly low zero-shot error on the held-out lab—slightly better than their average across the full test set—despite changes in background, lighting, and object distribution. Finetuning further improves accuracy and even surpasses a model trained from scratch in that environment, highlighting the value of pretrained dynamics even when embodiment remains constant. Simulation-pretrained models do not outperform scratch baselines, yet they reach comparable accuracy with $20\times$ fewer optimization steps, reinforcing the promise of simulation as a pretraining signal. Finally, the model pretrained on the combined DROID and BEHAVIOR-1K mix delivers mildly stronger zero-shot performance than DROID-only but offers similar finetuning gains.

### 5.4 MODEL-BASED PLANNING WITH POINTWORLD

An overarching goal of 3D world models is to generate actions that let robots operate autonomously in the real world. We therefore test whether a single PointWorld, pretrained across real and simulated interactions, provides enough interaction priors to enable zero-shot deployment on hardware. We integrate PointWorld into a model-predictive control loop and deploy on a Franka arm similar to DROID but with a different gripper morphology, explicitly probing morphology generalization.

For each task, we build a lightweight cost from an initial RGB-D capture: a manually selected mask and a target rigid transform specify the goal; for rigid and articulated objects we add small auxiliary terms (path length, workspace guidance). The action space is an end-effector pose trajectory. Each optimization rolls out 30 steps; because PointWorld predicts 10 steps jointly, we perform three autoregressive passes. Assuming quasi-static scenes (objects move only upon contact), we accelerate evaluation by filtering sampled trajectories: if a rollout has no predicted intersection with the scene point cloud, we hold the scene static and skip the model; only contact-inducing samples invoke forward passes.

With only the pretrained model and a shared MPC framework, PointWorld optimises actions for a broad set of real-world tasks: non-prehensile pushing of rigid objects (tissue box, book), deformable manipulation (folding a scarf, placing a pillow), articulated manipulation (opening a microwave, closing a drawer—revolute and prismatic), and tool use (sweeping with a duster or broom). These results indicate that PointWorld captures transferable interaction dynamics, including robot–object and object–object effects, despite differing gripper morphology at deployment.

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

# A    APPENDIX

## APPENDIX CONTENTS

## A.1    ADDITIONAL MODEL DETAILS

**Data pipeline.**    Synchronized RGB and depth from two calibrated cameras are fused into a partial 3D point set per timestep in the world frame. Using known camera extrinsics and joint configurations, pixels belonging to the robot are excluded when forming scene points. Each scene point stores position, color, surface attributes, and a distance-to-robot scalar. Supervision uses correspondences across timesteps; occluded or invalid-depth points are not used for loss.

**Feature extraction.**    Scene points are projected into each view and augmented with frozen DINOv3 ViT-L/16 features aggregated across scales and views (Siméoni et al., 2025). Robot points are sampled once at the first frame, attached to rigid links, and propagated to future steps by forward kinematics; they are projected to a shared channel with a linear layer and augmented with a sinusoidal time embedding and a learned robot-type embedding.

**Planning-time feature caching.**    For planning, scene features from the frozen encoder are computed once per observation and cached. During trajectory rollout, the dynamics operate on this fixed representation while chaining prediction windows across the horizon. This amortization reduces latency without altering model predictions, since the encoder is frozen, and it ensures consistent conditioning across all sampled trajectories.

**Backbone and heads.**    A custom Point Transformer V3 backbone ($\tilde{4}10M$ parameters; (Wu et al., 2023)) processes the unified point set and outputs features for scene tokens. A dynamics head predicts relative displacements to each future step; an uncertainty head predicts scalar log-variance per point and step. Predicted positions add relative displacements to input positions. The total parameter count is $\tilde{7}00M$ including the frozen DINOv3 encoder.

**Objective.**    The training loss is a Huber objective on residuals between predicted and ground-truth absolute positions (equivalently, relative displacements), applied in normalized coordinates. Per-point weights combine (i) soft movement likelihoods to emphasize moved regions and (ii) scalar aleatoric uncertainty to attenuate noisy supervision. Losses sum over predicted steps and valid correspondences only.

**Configuration.**    Context horizon $T_c{=}1$; prediction horizon $H{=}10$; per-step output normalization enabled; Huber delta 5.0; grid size 1.5 cm; maximum 12k scene points and 500 robot points per step (gripper-only); predictor width 256; patch size 256; drop path 0.3; dynamics head initialized near zero. Optimization uses AdamW with learning rate $1 \times 10^{-4}$ and weight decay $10^{-2}$; batch size 22 with distributed training.

## A.2    BACKBONE PROFILING METHODOLOGY

**Setup.**    Each backbone candidate is instantiated with the exact hyperparameters used in the main text experiments, including frozen feature encoders and point-cloud preprocessing. We evaluate using live batches drawn from the Droid training split, preserving the original two calibrated RGB-D views and all geometric augmentations. To keep comparisons on a single device, the per-GPU batch

Table 3: Training configuration summary.

| Setting | Value |
|---|---|
| Cameras per step | 2 (synchronized) |
| Context / Prediction horizons | $T_c=1$, $H=10$ |
| Scene / Robot points (max) | 12 000 / 500 per step (gripper-only) |
| Grid size | 1.5 cm |
| Depth threshold | 3 mm |
| Scene encoder | DINOv3 ViT-L/16 (frozen; multi-scale aggregation) |
| Backbone | Custom Point Transformer V3 ($\tilde{4}$10M params) |
| Predictor width | 256 |
| Regularization | Drop path 0.3 |
| Loss | Huber (delta 5.0); scalar aleatoric uncertainty |
| Movement weighting | Soft likelihood weighting of per-point loss |
| Normalization | Per-step output norm enabled |
| Batch size / Epochs | 22 / 200 |
| Optimizer | AdamW; LR $1 \times 10^{-4}$; WD $10^{-2}$ |

Table 4: Absolute profiling metrics for each backbone. Parameters include the frozen scene encoder; peak memory is the maximum device allocation during the profiled forward pass.

| Backbone | Total params (M) | Predictor params (M) | Peak mem. (GiB) | FLOPs ($10^{12}$) | Latency (ms) |
|---|---|---|---|---|---|
| GBND (baseline) | 1.37 | 1.03 | 1.87 | 0.17 | 41.0 |
| PointNet | 1.40 | 1.05 | 0.48 | 0.00 | 7.1 |
| PointNet++ | 1.44 | 1.10 | 1.28 | 0.01 | 317.0 |
| SPConv | 34.6 | 34.22 | 6.62 | 0.18 | 29.1 |
| Flat Transformer | 42.5 | 42.18 | 0.39 | 0.44 | 44.1 |
| PTv3 Small | 50.6 | 50.49 | 0.35 | 0.06 | 61.0 |
| PTv3 Medium | 130.9 | 130.70 | 0.67 | 0.11 | 75.3 |
| PTv3 Large | 409.9 | 409.58 | 1.86 | 0.30 | 119.7 |

size is capped at two sequences and data loading uses one worker per iterator. All measurements are performed on a single NVIDIA RTX 4090 (24 GB) with the remaining runtime identical across backbones.

**Procedure.** For every backbone we perform one warm start pass to populate caches, followed by a profiled forward pass through the full dynamics stack. Parameter counts are computed over the complete model (including the frozen scene encoder) and reported in millions. Peak device memory corresponds to the maximum allocated CUDA memory observed during the profiled run. FLOPs are obtained from the same profiling pass and reflect the total floating-point work for the batch, while latency records the elapsed kernel time for the forward call. No stochastic layers are disabled beyond the deterministic batch sampling described above.

**Results.** Table 4 lists the absolute measurements. Point-based alternatives such as PointNet and PointNet++ match the parameter footprint of the GBND baseline while reducing FLOPs, but they either increase peak memory or incur substantially higher latency. Hybrid volumetric architectures (SPConv, PTv3 variants) expand parameter count appreciably; the large PTv3 configuration surpasses 700M parameters and requires over 4 GiB for a single forward pass even with the reduced batch size.

A.3    DATASET CURATION AND EVALUATION DETAILS

REAL-WORLD ANNOTATION PIPELINE

**Inputs and calibration.** Each episode provides synchronized multi-view RGB streams with measured camera intrinsics (extrinsics are computed later), plus robot joint states and a kinematic model. We downsample time by a fixed ratio and align timestamps across cameras using the first camera as the canonical clock.

**Depth estimation and validity (step 1).** Per-view metric depth is obtained using a high-quality stereo estimator (Wen et al., 2025). Depth values are sanitized by clamping to a trusted range $[0, 4]$ m and producing a per-pixel validity mask. Per-point depth validity flags are propagated to 3D points.

**Camera extrinsics computation (step 2).** We do not use dataset-provided extrinsics. We compute camera extrinsics using a two-stage procedure that depends on accurate metric depth: (i) initialize multi-view poses via a transformer-based estimator (Wang et al., 2025) with either wrist- or external-camera anchoring; and (ii) jointly optimize 6-DoF updates for all external cameras using a robot-mesh, depth-alignment objective. The full formulation and defaults appear in Appendix A.3. This stage is crucial for achieving sub-centimeter alignment in the working volume.

**2D tracking and occlusion-aware seeding (step 3).** We form a per-view, per-clip tracking mask to avoid spurious correspondences: (i) a workspace mask obtained by projecting a fixed 3D workspace volume to the image; and (ii) a robot mask rendered by forward kinematics of the robot's URDF and projecting dense mesh samples to the image. A morphological closing fills small holes. The final mask is workspace and non-robot regions. We then track 2D points using strong point trackers (e.g., CoTracker (Karaev et al., 2023) or BootsTAP (Doersch et al., 2024)) on the masked regions only.

**3D reconstruction and multi-view fusion.** For each frame, valid depth, intrinsics, and extrinsics back-project tracked pixels to world-frame 3D points with RGB. Tracks across time yield temporally consistent per-point trajectories. We store per-camera trajectories to avoid mixing viewpoints prematurely and keep per-point visibility and depth-valid flags.

**Clip slicing and motion-based selection.** We slice into overlapping clips of length $F$ with stride $s$. Clips are retained if either the gripper changes state within the clip or end-effector motion exceeds thresholds. Thresholds depend on whether the gripper is predominantly open or closed within the clip: position/rotation thresholds are $(0.005\,\mathrm{m},\ 0.10\,\mathrm{rad})$ when open and $(0.002\,\mathrm{m},\ 0.05\,\mathrm{rad})$ when closed; either exceeding suffices to keep a clip. Gripper state changes always keep a clip.

**Outlier removal, smoothing, and normals.** We apply spatial outlier removal (density-based, typical $\varepsilon \in \{0.02, 0.05\}$ m, min-points 5 over a fraction of frames), followed by trajectory optimization with a multi-term objective (adherence, acceleration regularization, and surface-normal consistency), windowed in time. Finally, per-frame normals are estimated and oriented toward the camera, with a consistency fix to flip back-facing normals.

Table 5: Real-world annotation defaults.

| Setting | Value |
|---|---|
| Time downsampling | $10\times$ (canonical timestamps) |
| Clip length $F$ / stride $s$ | 16 / 1 |
| Depth trusted range | $[0, 4]$ m |
| Workspace bounds (relaxed) | $x \in [0.0, 0.7]$, $y \in [-0.4, 0.4]$, $z \in [-0.3, 1.2]$ m (with margin) |
| EE motion thresholds (open) | 0.005 m, 0.10 rad |
| EE motion thresholds (closed) | 0.002 m, 0.05 rad |
| Outlier removal | $\varepsilon \in \{0.02, 0.05\}$ m; min-pts $= 5$ |
| Trajectory smoothing | window $= 5$, $k{=}30$, radius $= 0.1$ m, max iter $= 50$, lr $= 10^{-4}$ |

CAMERA EXTRINSICS COMPUTATION

**Goal and inputs.** We compute camera extrinsics for real episodes to reduce metric errors from factory calibration and log drift. Inputs are: (i) timestamp-aligned multi-view RGB (including a hand-mounted wrist camera) with measured intrinsics; (ii) robot joint states and a kinematic/mesh model; and (iii) per-frame metric depth from stereo or a high-quality monocular model. Wrist images are rotated by $180°$ to match the world frame convention used during estimation.

**Initialization via multi-view pose estimation.** We obtain an initial, globally consistent set of camera poses using a multi-view transformer-based estimator (Wang et al., 2025). Two anchoring

strategies are supported: wrist-anchored, where the wrist view defines the reference frame and external cameras are averaged relative to the wrist across frames; and external-anchored, where the first external camera defines the reference and other cameras are averaged relative to it. Frame-wise pose averaging uses a standard SE(3) averaging operator. Accurate metric depth is required for the subsequent optimization stage. The fixed hand–camera transform is obtained from a per-robot calibration and composed with per-frame forward kinematics of the gripper pose to map between base and wrist frames.

**Robot-visibility filtering.** We filter frames to retain only those where every external camera observes a sufficient number of robot surface points. For each frame, dense robot mesh points are generated via forward kinematics, transformed to each camera, and projected with measured intrinsics. We then count points with valid depth support in the selected depth source (within a trusted range). Frames where any camera sees fewer than a threshold (e.g., 2000 points) are removed before refinement.

**Joint optimization objective.** We refine all external cameras jointly by optimizing a small 6-DoF perturbation per camera about the initialization (translation/rotation scales set to centimeters/degrees). Let $\mathbf{X}$ be robot surface points in world coordinates, $\Pi_K(\cdot)$ the camera projection with intrinsics $K$, $T_c$ the camera pose in world coordinates, and $D$ the observed depth map. For each frame and camera we define the per-camera loss as the mean absolute depth residual between the predicted camera-space depth $z(T_c^{-1}\mathbf{x})$ and the observed depth sampled at the projected pixel, restricted to pixels within image bounds and valid depth range, with light deduplication of projected coordinates:

$$\mathcal{L}_c = \text{mean} \left| D\big(\Pi_K(T_c^{-1}\mathbf{x})\big) - z(T_c^{-1}\mathbf{x})\right|.$$

The total objective averages $\mathcal{L}_c$ across all retained frames and cameras. We use first-order optimization (Adam) on the 6-DoF updates with typical defaults (learning rate $10^{-3}$, up to 100 iterations), and report per-camera translation/rotation deltas and robot-point support.

**Depth sources and ranges.** Depth can be sourced from synchronized stereo (preferred) or from a learned model; in both cases we enforce a trusted range (e.g., $[0.3, 2.0]$ m) during residual computation. Intrinsics are paired to the chosen depth source.

**Outputs.** The procedure returns refined extrinsics for all external cameras in the robot base frame, along with diagnostics: initial/final data loss, average/min/max robot point counts per camera-frame, and per-camera SE(3) deviations from the initialization. We optionally produce multi-frame point-cloud visualizations colored by camera to inspect geometric consistency in the working volume.

Table 6: Extrinsics computation defaults.

| Setting | Value |
| --- | --- |
| Anchor | Wrist or first external camera |
| Min robot points (per camera-frame) | 2000 |
| Depth range (residuals) | $[0.3, 2.0]$ m |
| Deduplication threshold | 0.5 px |
| Optimizer / iterations | Adam / 100 |
| Learning rate | $1 \times 10^{-3}$ |
| Update scales | $\sim$1 cm (t), $\sim$1° (r) |

SIMULATION DATA GENERATION

**Environment and sensors.** We replay bimanual manipulation episodes in a physically realistic simulator with three fixed external RGB–D sensors (left, right, head). Image size is typically $256 \times 320$ with known intrinsics and poses. We enable high-quality ray-traced rendering and collect per-pixel depth, normals, and instance IDs.

**Per-camera 3D and object decomposition.** Each sensor back-projects valid depth to world points and corrects normals toward the camera. Using instance segmentation, we group points by USD prim and record, per visible object: (i) a set of local-frame surface points; (ii) the object's 7-DoF pose at

the clip start; and (iii) a per-frame rigid trajectory in the robot base frame. Storing local points plus rigid trajectories enables exact reconstruction in any frame without scale ambiguity.

**Clips and validity.** We form clips with temporal downsampling and stride. A clip is kept if it passes collision and transition checks and satisfies at least one of: (1) some object moves and non-base robot joints move; (2) some object moves concurrently with finger contacts; or (3) no objects move but the gripper and arm move (negative samples). End-effector thresholds mirror the real-world policy but use larger margins typical for simulation (e.g., $0.20$ m, $90°$ when open; $0.10$ m, $45°$ when closed). Workspace bounds are enforced in the robot frame.

**Robot signals.** For each arm we store per-frame gripper pose, openness, grasp state, and full joint positions. We also maintain world $\leftrightarrow$ robot transforms at the clip start to express all camera extrinsics and object trajectories in a common robot frame.

Table 7: Simulation extraction defaults.

| Setting | Value |
|---|---|
| Image size | $256 \times 320$ |
| Temporal skip / clip length / stride | 6 / 11 / 5 |
| Object movement thresholds | $0.05$ m, $45°$ |
| EE thresholds (open / closed) | $0.20$ m, $90°$ / $0.10$ m, $45°$ |
| Workspace bounds (robot frame) | $x \in [0.0, 1.3]$, $y \in [-0.8, 0.8]$, $z \in [-0.05, 2.0]$ m |

EVALUATION PROTOCOL

**Primary metrics.** Evaluation aggregates per-sequence losses and reports dataset- and domain-level summaries. The primary error is a Huber loss on absolute 3D positions between predictions and ground truth over all predicted steps, masked by (i) scene existence, (ii) non-context frames, and (iii) valid correspondences (occluded or invalid-depth points are excluded). We also track per-frame counts of available scene points, robot points, and supervised points.

**Expert confidence filtering (real domains).** For real domains, we optionally compute expert low-confidence masks by voxelizing world points with grid size $g$ and marking voxels deemed unreliable. During evaluation, a per-point confidence $c \in \{0, 1\}$ is decoded at each timestep and applied as an additional mask, yielding both unfiltered and filtered metrics. We record the fraction of supervised points kept by the confidence filter and expose per-domain aggregates.

A.4 DATA AUGMENTATION

Standard point-cloud and color augmentations are applied while preserving calibration consistency between 2D features and 3D points. Table 8 summarizes typical settings.

Table 8: Data augmentations (typical settings).

| Augmentation | Setting |
|---|---|
| Spatial scaling | Uniform in $[0.9, 1.1]$ |
| Axis flips | Probability 0.5 |
| Chromatic jitter | Probability 0.95; std 0.02 |
| Chromatic translation | Probability 0.95; ratio 0.02 |
| Auto contrast | Probability 0.2; blend 0.2 |
| Per-point 3D jitter | Std $0.001$ m; clip $0.003$ m |
| Sphere cropping | Enabled; radius in $[0.1, 0.8]$ m; buffer $0.25$ m |

A.5 MODEL-BASED PLANNING DETAILS

**Robot embodiment and sensing.** Experiments use a 7-DoF arm with a parallel gripper. The end-effector action parameterization is absolute $[x, y, z, \mathrm{roll}, \mathrm{pitch}, \mathrm{yaw}, \mathrm{open}]$. Gripper openness

is normalized using URDF joint limits. Two calibrated RGB-D cameras provide synchronized observations; camera intrinsics and extrinsics are applied to fuse a partial scene point set per timestep.

**Planner.** The planner samples action sequences around a nominal using cubic splines with $n_{\text{knots}} = 4$ and degree 3. Noise scales are scheduled between $\sigma_{\min}=0.05$ and $\sigma_{\max}=0.50$. Each refinement iteration draws 256 samples; importance weights use temperature $\beta=0.05$, and the nominal is updated with an exponential moving average (EMA = 0.9). The horizon for open-loop planning is $H=30$ steps with a one-step context; the horizon is chunked to match the prediction window of the dynamics model.

**Costs.** Task cost is specified over the predicted point-flow trajectory of the scene, enabling broad applicability across rigid, deformable, articulated, and tool-use tasks; concrete task instantiations are presented in the Experiments. Control costs include a piecewise-linear SE(3) path-length penalty (translation and rotation terms) and a reachability term that uses an inverse-kinematics residual as a proxy. A terminal-step scaling factor emphasizes goal satisfaction at the last step of the horizon.

Table 9: Planner configuration and bounds.

| Setting | Value |
|---|---|
| Samples per iter | 256 |
| Horizon | 30 steps (context 1) |
| Spline noise | Knots 4, degree 3, $\sigma \in [0.05, 0.50]$ |
| Importance weights | $\beta = 0.05$, EMA 0.9 |
| Workspace bounds | $x \in [-0.8, 0.2]$, $y \in [-0.5, 0.5]$, $z \in [0.0, 0.6]$ m; RPY relative bounds as specified |
| Regularization | SE(3) path length; IK reachability residual |
| Task terms | Subset tracking; push-approach side pose |
| Terminal scaling | Enabled on final step |

### A.6 LLM USAGE

Large language models (LLMs) were only used as supplementary tools to enhance the text by refining phrasing, correcting minor typographical errors, and improving the overall clarity of the paper.

