# OpenReview forum: "PointWorld: Scaling 3D World Models for In-The-Wild Robotic Manipulation"
_ICLR.cc/2026/Conference — ICLR 2026 Conference Withdrawn Submission_

### Official Review · Reviewer_xu3Q · 2025-10-29

**Soundness:** 3
**Presentation:** 2
**Contribution:** 2
**Rating:** 2
**Confidence:** 4

**Summary:**

This paper introduces Point World, a foundation 3D world model that predicts dynamics for robotic manipulation by unifying scene state (full-scene 3D point cloud from RGB-D) and robot action (dense robot point trajectories) in a shared 3D spatial domain. The model is trained to predict short-horizon 3D point flow (displacement). To enable this, the authors curate a large-scale, high-quality dataset (~2M trajectories, 500 hours) by combining real-world (DROID) and simulated (BEHAVIOR-1K) data, leveraging a new pipeline to produce dense 3D annotations for the real data. Through systematic scaling studies, they establish key design principles (e.g., using PTv3 and DINOv3 features). Finally, they demonstrate the model's ability to power zero-shot Model Predictive Control (MPC) on real hardware for diverse tasks, including rigid, deformable, and articulated manipulation, and tool use.

**Strengths:**

- The paper conducts a rigorous and systematic empirical study on design choices (backbone, loss function, features, and scaling).
- The careful curation and open-sourcing of the large-scale 3D dynamics dataset is a substantial, high-quality technical contribution.
- The paper is well-structured and the key components of the method and experiments are clearly explained. The use of tables and figures (e.g., the scaling roadmap in Figure 4 and the backbone comparisons in Table 1) effectively conveys the results.

**Weaknesses:**

- **Lack of Clear Motivation and Comparative Advantage**: The paper does not sufficiently articulate the clear advantage of this specific method compared to other large-scale dynamics or control approaches. A clearer motivation is needed to explicitly demonstrate why this 3D point-flow formulation is superior to established alternatives.
- Lack of Downstream Control Baselines: While the paper provides architectural ablations for the predictive task (Table 1), the crucial zero-shot model-based planning results lack a comparison against contemporary competitive baselines. Success rates (e.g., 70-80%) are presented in isolation, making it difficult to judge PointWorld's practical efficiency and superiority over other large-scale visual dynamics models or model-free approaches that could potentially leverage the same data mixture for similar zero-shot performance.
- Incomplete Document Structure: The paper lacks the standard dedicated final section for "Conclusion and Future Work", which is typically expected for a complete academic paper to fully summarize findings and formally outline research directions.

**Questions:**

- Inference Speed/Latency: Given the scale of the PTv3 backbone (411M parameters) and its use in a Model Predictive Control (MPC) loop, what is the inference latency (in milliseconds) for a single 10-step prediction pass on the deployed hardware?
- Robot Embodiment Generalization: The abstract claims generalization to unseen robot embodiments. Can the authors provide specific quantitative results (e.g., success rates or prediction error) on a third, completely different type of physical robot arm (e.g., one with a different kinematic structure than the two primary robots used in the DROID/BEHAVIOR datasets) to robustly demonstrate the generalization capacity across robot kinematic structures?

---

### Official Review · Reviewer_Uew4 · 2025-11-01

**Soundness:** 3
**Presentation:** 3
**Contribution:** 2
**Rating:** 6
**Confidence:** 3

**Summary:**

This paper introduces PointWorld, an action-conditioned 3D world model that unifies states and actions in a shared point-cloud space. By predicting per-point 3D displacements, it captures dynamic interactions in both real and simulated environments, enabling generalized physical prediction across diverse scenes. The authors further present a large-scale 3D interaction dataset (2 M trajectories, 500 hours) spanning multiple robot morphologies and sim-to-real domains, and systematically study architecture, objectives, and scaling. Experiments show strong zero-shot generalization on real-world manipulation tasks, including rigid, deformable, articulated, and tool-use cases—demonstrating that PointWorld can infer scene dynamics from limited RGB-D inputs and support real-robot control without task-specific training.

**Strengths:**

- The paper proposes a world-modeling paradigm that represents both state and action in a shared point-cloud space, avoiding the limitations of low-dimensional physics simulators or voxel/mesh state encodings. Mapping RGB-D perception and robot kinematics to a joint point-flow prediction task yields a physically intuitive and highly extensible formulation.
- The proposed metric-stereo depth estimation, automatic extrinsic calibration, and marker-free tracking pipeline address the long-standing issue of accurate 3D supervision in real scenes. Coupled with BEHAVIOR-1K simulation data, it forms a unified sim-to-real training corpus.
- The paper thoroughly evaluates the effects of architecture (GBND→PTv3), objective design (Huber + uncertainty + motion weighting), pretrained feature backbone (DINOv3), and model size. The results reveal clear scaling-law trends—demonstrating predictable performance growth with data and capacity.
- The model achieves zero-shot MPC control on the Franka arm across rigid, deformable, articulated, and tool-manipulation tasks, confirming robust cross-morphology and cross-scene generalization.

**Weaknesses:**

- The paper proposes unifying states and actions within a 3D point-space representation, but it does not provide a clear physical or dynamical justification for this formulation. Equation (1) treats 3D point-flow prediction largely as a black-box function approximation, without discussing its theoretical advantages or stability properties.
- The training objective combines Huber loss, motion weighting, and aleatoric uncertainty, yet the independent contribution and weighting strategy of each component are not ablated, leaving the necessity and optimality of this design insufficiently supported.
- Moreover, the paper lacks direct quantitative or conceptual comparisons with recent world-model frameworks, which weakens the articulation of PointWorld’s unique advantages. A deeper analysis contrasting its representational or performance characteristics with these methods would strengthen the paper’s contribution.

**Questions:**

Please see weakness.

---

### Official Review · Reviewer_VbZW · 2025-11-02

**Soundness:** 3
**Presentation:** 3
**Contribution:** 3
**Rating:** 4
**Confidence:** 4

**Summary:**

This paper introduces PointWorld Model – A foundation 3D world model predicting full-scene point flow from RGB-D + actions. PointWorld bridges the gap between 3D perception and physical interaction. It provides unifies perception, dynamics, and action in a common 3D domain — elegant and embodiment-agnostic. It also provides massive 3D Dynamics Dataset – 2M trajectories (real + simulated) with high-quality metric depth and correspondences.

There are also some concerns and limitations, especially for robotics experiments. 1. The pipeline to extract robotic action lack of the capability for challenging embodiment, e.g. dexterous hands, mobile arm ans so on. 2. Rely on imagination, but might not be perfect for action alignment and physically accuracy. 3. Rely on pre-existing 3d depth recovery, which might cause some errors.

**Strengths:**

1. PointWorld establishes a unified and scalable framework for 3D world modeling in robotic manipulation.
2. Introduces large 3D dynamics dataset (≈2M trajectories, 500 h) combining real DROID data and BEHAVIOR-1K simulation with precise metric depth and correspondences.
3. Demonstrates that large-scale 3D world models can act as foundation models bridging perception, physics, and control.
4. Enables model-predictive planning on real hardware for rigid, deformable, articulated, and tool-use tasks without task-specific fine-tuning.

**Weaknesses:**

1. No comparison with other robotic foundation models on zero-shot generalization capability
2. The pipeline to extract robotic action lack of the capability for challenging embodiment, e.g. dexterous hands, mobile arm ans so on.
3. The world model rely on imagination, but might not be perfect for action alignment and physically accuracy.
4. Rely on pre-existing 3d depth recovery, which might cause some errors.
5. This method can only serve for static camera, for dynamic camera, it cannot handle camera movement and 3d reconstruction scope.

**Questions:**

See weakness.

---

### Note · Authors · 2025-11-14

I have read and agree with the venue's withdrawal policy on behalf of myself and my co-authors.